# Computational, *In Vitro,* and *In Vivo* Models for Nose-to-Brain Drug Delivery Studies

**DOI:** 10.3390/biomedicines11082198

**Published:** 2023-08-04

**Authors:** Radka Boyuklieva, Plamen Zagorchev, Bissera Pilicheva

**Affiliations:** 1Department of Pharmaceutical Sciences, Faculty of Pharmacy, Medical University of Plovdiv, 4002 Plovdiv, Bulgaria; radka.boyuklieva@phd.mu-plovdiv.bg; 2Research Institute, Medical University of Plovdiv, 4002 Plovdiv, Bulgaria; plamen.zagorchev@mu-plovdiv.bg; 3Department of Medical Physics and Biophysics, Faculty of Pharmacy, Medical University of Plovdiv, 4002 Plovdiv, Bulgaria

**Keywords:** nose-to-brain delivery, computational models, *in vitro* models, *ex vivo* models, *in vivo* models

## Abstract

Direct nose-to-brain drug delivery offers the opportunity to treat central nervous system disorders more effectively due to the possibility of drug molecules reaching the brain without passing through the blood–brain barrier. Such a delivery route allows the desired anatomic site to be reached while ensuring drug effectiveness, minimizing side effects, and limiting drug losses and degradation. However, the absorption of intranasally administered entities is a complex process that considerably depends on the interplay between the characteristics of the drug delivery systems and the nasal mucosa. Various preclinical models (*in silico*, *in vitro*, *ex vivo*, and *in vivo*) are used to study the transport of drugs after intranasal administration. The present review article attempts to summarize the different computational and experimental models used so far to investigate the direct delivery of therapeutic agents or colloidal carriers from the nasal cavity to the brain tissue. Moreover, it provides a critical evaluation of the data available from different studies and identifies the advantages and disadvantages of each model.

## 1. Introduction

Over the past few decades, the hypothesis that after intranasal administration drug molecules and colloidal drug carriers can reach the central nervous system (CNS) (i.e., nose-to-brain delivery) bypassing the blood–brain barrier (BBB) has received much attention [1]. In fact, this is possible thanks to the olfactory mucosa that provides a direct connection between the external environment and the central nervous system through the olfactory nerves. The BBB is known to be the main barrier to the access of xenobiotics to the cerebrum. Through the nasal route of administration drug molecules can directly reach the brain, avoiding enzymatic and chemical destruction processes, as well as hepatic first-pass metabolism. The nasal mucosa is richly blood-supplied with good permeation properties that provide rapid drug absorption [2]. Different dosage forms can be easily self-administered intranasally without pain and the requirement of specific techniques [3]. Therefore, nose-to-brain delivery represents a great opportunity to deliver both conventional drugs and biotherapeutics for the treatment of CNS disorders, such as Parkinson’s disease (PD) [4,5] and Alzheimer’s disease (AD) [6] while avoiding the use of invasive techniques to reach the brain.

After the intranasal administration of drugs and biopharmaceuticals, problems may arise due to their physicochemical properties (e.g., molecular weight, solubility, stability). On the other hand, rapid clearance of the formulation may occur due to the typical anatomical characteristics of the nasal cavity.

In this regard, the development of effective dosage forms is necessary. Formulation strategies can combine micro- and nanotechnologies and the use of excipients extending nasal residence time. In addition to formulation characteristics, nasal delivery requires a device to adequately aerosolize the dosage form and ensure its deposition in the required area of the nasal cavity. The formulation, including its excipients along with the dosing device should be thoroughly characterized to explicate both excipient and device choice [2]. 

Experimental models allow for preclinical investigation of the administered drug, which will further be used to correlate the brain-targeting potential of intranasal administration in humans. *In silico*, *in vitro*, and *in vivo* intranasal models represent a tool to investigate pharmaceutical and physiological factors that may play different roles in drug delivery across the nasal mucosa, as well as to determine the mechanisms involved in drug absorption from the nasal cavity. Therefore, it is of crucial importance to determine the actual site and extent of drug deposition. For this purpose, computational models, *in vitro* tests, and *in vivo* experiments have been developed and used.

*In silico* methods are mainly used to predict drug pharmacokinetics and to determine key parameters, and how they are affected by the drug molecular size, solubility, pKa, and drug release behavior [7].

Cell-based *in vitro* studies of intranasal drug delivery involve the use of various monolayers of epithelial cells. These cell-based models can be used to study the mechanisms of drug transport across epithelium, e.g., transcellular and paracellular passive diffusion, active transport, and active efflux transport, meanwhile supporting studies where the mechanism of action is investigated. However, some disadvantages of cell-based models can be outlined, mainly due to variations in culture conditions, the level of transport molecules expression, or the absence of a mucous layer, leading to considerable intra-laboratory variability [8].

*Ex vivo* studies refer to the application of living tissues for experimental research outside the intact organism. The term can sometimes be used synonymously with “*in vitro*” (in glass—meaning in a test tube). For this review article, *ex vivo* models refer to nasal tissues collected from different animals for the purpose of studying different aspects of nasal absorption of drugs in a standardized environment outside of an intact living organism to the usage of different tissues instead of the entire organisms for experimental studies. The term is synonym to “*in vitro*”, i.e., in glass or in a test tube. In this review article, *ex vivo* models based on animal nasal tissues are involved to study different factors affecting nasal absorption of drugs. Using such models enables us to obtain relevant data on the pharmacokinetic behavior including absorption, metabolism, or elimination, as well as tissue toxicity [9]. 

*In vivo* models play a crucial role in evaluating drug bioavailability and time course of drug from the nasal cavity to the central nervous system [10]. *In vitro* and *ex vivo* models are not sufficient to recreate the internal environment of the nasal mucosa and the olfactory nose-to-brain passage.

Although there is a wide range of preclinical methodologies used to evaluate the transport of drugs and colloidal carriers across the nasal mucosa, each model has its own advantages and disadvantages (summarized in Table 1).

This review aims to provide an overview of computational, *in vitro*, and *in vivo* models that have been developed to study direct nose-to-brain drug delivery.

## 2. Computational Analyses

Computational analyses are used to numerically study the behavior of complex systems by means of a computer simulation. They can be used to predict the system behavior under various conditions, often for cases in which intuitive analytical solutions are not available. In the last few years, there has been a widespread interest in computational technologies in both academia and pharmaceutics [11]. Computational models can aid the rational design of new and safe drug molecules, and their incorporation into appropriate drug delivery systems, limiting the use of animal models in pharmacological research [12,13,14]. Computational approaches to analyze the delivery of drugs from the nose to the brain through the olfactory region are intended to simulate the airflow during inhalation, specifying the movement of gases, droplets, particles, and their deposition on the olfactory mucosa. There is a substantial requirement for computational models to depict the interaction of deposited objects with the mucus layer, drug release, and transfer across the mucus layer. 

### 2.1. Computational Fluid Dynamics (CFD) Models for Drug Deposition

Computational fluid dynamics (CFD) is a branch of fluid mechanics that uses applied mathematics, physics, and computational software to visualize how a gas or liquid flows, as well as how the gas or liquid affects objects as they flow past them. Computational fluid dynamics (CFD) is a science, part of fluid mechanics, that simulates and analyses gases or liquids’ flow and how they act on objects that flow past them. Different simulations and their analysis are based on applied mathematics, physics, and computational software. The most common CFD tools are based on the Navier–Stokes equations which define single-phase (gas or liquid, but not both) fluid flows [15]. It is a feasible tool for investigating and troubleshooting different types of equipment used in the pharmaceutical industry or healthcare. CFD can increase the efficiency of device development and reduce the need for expensive and time-consuming laboratory experiments.

CFD is mostly used to elucidate the basis of airflow in the nasal cavity and the design of the nose-to-brain drug delivery system after subsequent drug transfer and olfactory deposition. Additionally, it has been used to monitor and record important data such as nanoparticle trajectories and deposition sites. The delivery of particles to the olfactory region of the nasal cavity is a major challenge due to the complex geometry of the nasal cavity—highly curved nasal passages, differences in nasal structure from person to person, and dynamic airflow, such as a rapid transition from laminar to turbulent. Current nasal delivery devices such as droppers, nasal sprays, and atomizers have some drawbacks that could be upgraded. Automated inhaler devices such as dry powder inhalers (DPIs) or assisted devices—metered dose inhalers, mDIs—are used to deliver solid or liquid dosage forms to the olfactory region of the nasal cavity [16,17,18]. The configuration of the air passages, the turbulence of the airflow and the physicochemical properties of the particulates—size, shape, density, and hygroscopicity—are the major factors that affect the removal of particles from the airflow and their penetration depth into the air passages [19]. The angle of the spray nozzle piece in the nasal vestibule and the velocity of the particles from the nozzle also contribute. The linear velocity, the turns in the passage, and the degree of turbulence of flow are the most important characteristics of the airflow. Gravity and inertia also affect particle deposition. The effect of gravity is proportional to the mass of the particles and affects their tendency to settle. Inertial forces will affect the fate of a particle in proportion to both its mass and the velocity of the flow in which it is suspended. This force is of particular importance in relation to airflow curves caused either by a change in the direction of the air passage or by turbulent flow. 

To perform successful CFD calculations of the nose, it is necessary to construct an appropriate surface mesh of the nasal cavity. The surface mesh should then be imported into a volume mesh generator and finally the simulations will be carried out by commercial CFD code. Cell quality should be tested during surface and volume mesh generation. CFD simulations of the nasal cavity airflow are mostly performed with the k-ω SST RANS approach or Large Eddy Simulations, LES, which are very computationally expensive.

Kleven et al. described how the Norwegian company OptiNose AS used CFD during the development process of their bi-directional delivery system. Exhalation into the delivery device triggers the release of particles into an air stream that enters one nostril through a sealing nozzle and exits through the other nostril. This is possible due to the posterior connection between the nasal passages persisting when the velum closes automatically during oral exhalation. They used the CFD simulations to visualize and demonstrate the key features of the bi-directional delivery system that improves drug and vaccine distribution in the nasal mucosa while preventing deposition in the lungs. After acquiring high-resolution computed tomography (CT), they created a surface mesh with nasal geometry, a volume mesh, and then ran simulations of the nasal cavity using the commercial CFD code FLUENT. They applied the Discrete Phase Model (DPM) which follows the Euler–Lagrange approach. The gas phase was treated as a continuum by solving the Navier–Stokes equations. The airflow was simulated as a laminar flow. The particle deposition rates increased with both increasing air velocity and increasing particle size. By increasing the particle size but maintaining constant air velocity due to manifestation of inertial forces, so fewer particles follow the air stream around the nasal septum and exit through the other nostril. When particles with a uniform size distribution around 10 µm were used, 30.8% of the particles circumvented the bi-directional nasal passage, in contrast to 3.5 µm particles, which demonstrated 54.6% bypass. The results were validated by comparison with a physical experiment with gamma-scintigraphy studies, which were promising. The gamma scintigraphy result was 36.9 ± 18.4% for 10 µm particle size and 22 ± 7.5% for 3.5 µm particle size. For direct comparison with the FLUENT simulation, the subject that was CT-scanned for this model had 23.9% and 19% in the gamma study, respectively. Consequently, FLUENT simulations underestimate particle deposition in the nasal cavity [20]. Considerably higher deposition of microparticles in the olfactory region was observed with the bi-directional delivery technique compared to conventional inhalation. Particles 14 µm to 18 µm in size can increase olfactory deposition when introduced with bi-directional delivery technique. 

Xi et al. validated their experimental results for a bi-directional nasal drug delivery system numerically. They established that the bi-directional delivery technique enhanced the deposition of particles in the upper part of the nasal cavity, especially in the olfactory region [21,22]. 

Targeted aerosol delivery to the olfactory region with bi-directional pulsatile flow conditions was analyzed by Farnoud et al. They used a CFD model on patient-specific nasal geometry. Computational aerosol deposition studies were interpreted through deposition efficiency (DE) and locally deposited drug—the mass per unit area. The study demonstrated that regardless of the low DE (<1%) in the olfactory region, the local drug amount in this region did not differ from the rest of the nasal cavity due to the fact that the olfactory region covers a 64-fold smaller area. [23]. The modelling accuracy was proved with experiments on generic bend geometries and real measurements of concrete nasal aerosol deposition in a normal flow.

Ren et al. used CFD to simulate the effect of auxiliary airflow on the deposition of drug particles in the olfactory region during nasal administration. Firstly, they created a realistic 3D nasal model based on a CT scan of the head of a healthy adult. Particles’ release from a particular site was simulated by CFD-DPM model. Various operating conditions were used for the analysis of flow field and the fraction of deposited particles. The effect of particle size on the deposition fraction in the olfactory region varied significantly with different auxiliary flows. A delivery velocity of 15 m/s was found to be more favorable for nasal administration. A spray cone angle of 40° was more suitable for drug delivery at a lower auxiliary airflow, while a spray cone angle of 60° was more efficient at a higher auxiliary airflow. To verify the accuracy of the nasal cavity model and method, an experimental study of drug particles deposition was carried out. The simulation results were in acceptable agreement with the experimental results. There were some errors in the results from different regions, but they were within reasonable limits so that the model and method could obtain reliable results [24].

So far, computational simulations have revealed small deposits near the olfactory region, varying with flow velocity and particle size. The greatest deposition rate was observed for fine particles, as larger particles are lost due to inertial collision or following the main airflow paths. For fine particles, the deposition increases with decreasing particle size due to diffusive movement of the particles. The lack of satisfactory validation of these models and the fact that they cannot be directly related to therapeutic outcomes hinder their immediate clinical application. Furthermore, various variables affect spray deposition, and since there is no single factor associated with specific nasal deposition, empirical studies are essential to validate targeted drug delivery [25].

### 2.2. Carrier/Mucosa Interaction Models

Different interactions can occur between drug delivery systems and the olfactory mucus layer due to the physical state of the dosage form. Liquid droplets may spread to some extent, coalesce, or separate from mucus layer. A transferred drug model is needed to describe the diffusion of drug molecules to the mucus interface, and the partition between the liquid and mucus phase. Kiparissides et al. modeled the deposition of a droplet of hyaluronic acid on a mucus substrate by a dynamic droplet deformation model. They then formulated a dynamic drug release model to quantify the release rate of an active pharmaceutical ingredient. Finally, they modeled drug transport from the hydrogel–mucosa interface to the olfactory bulb through the olfactory sublayers of the epithelium and lamina propria with a series of dynamic mass transport models. They integrated different mathematical models together, following a multi-scale modeling approach to help identify key design system parameters and material properties that can lead to formulation optimization [26]. 

Solid drug delivery systems can spread over the mucus layer and pass through it or stay on the mucosal surface somewhat fixed in the mucus. Drug carrier deposition is followed by mass exchange and drug release. More than one model is required to describe all possible interactions with the mucosal layer. Models describing adhesion to the mucosal layer can reveal attachment stability along with chain interpenetration, electrostatic, and hydrophobic interactions, dispersion forces, etc. [27]. Among *in silico* modeling of different variables, drug release studies are crucial for accurate computations. The manner of drug release from particulate carriers should be investigated in terms of mucus layer, particle swelling performance, biodegradability, and deposition area. The rate of drug release from the carriers is of great importance. Computational studies of drug release from different formulations have been performed for a variety of drugs. The amount of released substance should be evaluated under various environmental conditions in terms of temperature, humidity, lipophilicity, and ionic strength. Unfortunately, there are not many articles on drug release patterns in the olfactory area of the nasal cavity. Kiparissides et al. developed a dynamic drug release model to describe the release rate from microcarriers embedded in a hydrogel matrix. After numerical solution of various computational models, the dynamic mass transfer rate of the drug from the hydrogel matrix to the mucosal interface was calculated. Drug transfer from the hydrogel–mucosa interface to the olfactory bulb was also calculated [28]. However, the real systems are much more complicated and morphological changes can occur within the particle during degradation, significantly influencing the drug release rate.

### 2.3. Mucus Layer Penetration Models

The mucus layer is a complex biological dispersion system containing water in which the glycoprotein mucin, enzymes, lipids, macromolecules, macrophages, etc., are dispersed. Particulate systems or drug molecules can interact with mucin chains or other dispersed entities, which can significantly affect their transport through the mucus layer. Computational simulations are limited by computational costs when it is necessary to describe the interactions between drug molecules or particles and the biological environment. Many drug molecules diffuse with characteristic times of microseconds and nanoparticles (NPs) with milliseconds. Brownian and Stokes dynamics can describe diffusion processes well, but they usually ignore intramolecular and conformation dynamics [29]. A full molecular dynamics simulation is not sufficient to describe the mucus layer due to its heterogenicity. Coarse-grained molecular dynamics (CGMD) has been used to represent the mucus layer at the atomic level [30,31]. The CGMD approach can be used for the prediction and description of many processes, e.g., hydrodynamic interactions, electrostatic interactions, and hydrophobic effects on microsecond to millisecond time scales [29]. 

## 3. *In Vitro* Methods

*In vitro* methods were developed to replace *in vivo* and *ex vivo* methods. A variety of cell models are available to study intranasal drug delivery which include cell lines and immortalized cell cultures [32]. Their aim is to mimic the human nasal mucosa and should cover all physiological properties that may affect the drug pharmacokinetics after intranasal administration [33]. Appropriate cell lines must be selected to reproduce economically viable results. There are several *in vitro* cell culture models such as BT (normal bovine turbinate), NAS2BL (rat nasal squamous carcinoma), RPMI 2650 (human nasal epithelium), 16HBE14o- (human normal bronchial epithelium), Caco-2 (human colon carcinoma), Calu-3 (human lung adenocarcinoma), etc. They can be used to study the membrane permeability of drug molecules or colloidal carriers. Cell models offer experimental control of the culture conditions—growth and differentiation of epithelial cells, or high-throughput screening. They can elucidate drug transport mechanisms and investigate the effect of absorption-enhancing methods on drug delivery [34].

Primary cell cultures and immortalized cell lines are increasingly used as *in vitro* models to assess nasal permeability (Table 2). Human cell cultures are preferred over animal cells because their similarity is greater, and they are clinically relevant. Furthermore, compared to *in vivo* models, they require a smaller drug amount to be tested due to animal dimensions and do not have ethical and regulatory issues as in animal models [35]. However, when using cell-based models, a risk of interlaboratory variability must be considered due to differences in culture conditions, laboratory procedures, varying expression of certain transporters, lack of a mucus layer, etc. [36]. 

### 3.1. Primary Cell Cultures 

Primary cells are collected from donors (living organisms such as humans, rats, or pigs) and then cultured under *in vitro* conditions. However, it is rare to obtain sufficiently reliable human cells from a single donor. Usually, several samples from different individuals are used and this leads to heterogeneity between cell cultures due to donor-to-donor variability. The anatomical and physiological characteristics of the nasal cavity must be considered—the type, number, and density of cells, as well as the presence of microvilli, as they significantly affect permeability. Sampling methods are divided into two groups—traumatic and atraumatic/non-surgical. Primary cell cultures are obtained when turbinectomy, nasal reconstitution, nasal polyp, or hyperplasic concha surgery is performed, all of which are traumatic techniques. Large amounts of cells can be collected with various applications, but it is difficult to repeat. Atraumatic/non-surgical methods such as nasal brushing, nasal swabs, lavage, and blown secretion are also used, but fewer cells are collected. However, atraumatic methods do not require anesthesia, allow multiple sampling and isolation of cells from the same source, and do not require proteolytic enzymes [32]. 

Rat olfactory mucosa is commonly used to assess nose-to-brain delivery. Specific criteria for the establishment of olfactory mucosa are defined: inoculation density >5 × 10^5^ cells/insert (0.9 cm^2^) as well as transepithelial electrical resistance (TEER) values > 160 Ω.cm^2^ after 21 days of culturing. The extracted cells are confirmed to be olfactory cells due to the expression of 5-AC mucin [47]. Gartziandia et al. developed, characterized, and validated *in vitro* olfactory cell monolayers from nasal cavity of Wistar rats to study nanoparticle (NP) transport across them. Poly(lactide-co-glycolide) (PLGA) and nanostructured lipid carrier (NLC) formulations were biocompatible with the olfactory mucosa cells. Although only 0.7% of PLGA NPs were able to pass across the olfactory cell monolayers, 8% and 22% of NLC and chitosan-coated NLC were transported across them, respectively. They found that the incorporation of cell-penetrating peptides to the NLC surface resulted in an increase in their transport by up to 46% [48]. Musumeci et al. developed poly-lactic acid (PLA), PLGA, and chitosan NPs loaded with fluorescent marker rhodamine. Their aim was to assess the uptake process of different types of NPs on olfactory ensheathing cells (OECs) by confocal microscopy after 1, 2, and 4 h. They also employed surface ζ-potential measurements to study the interactions between nanoparticles and cells. OECs were isolated from 2-day-old rat pups. The results showed that the uptake of rhodamine-loaded NPs by the OECs was time dependent and PLGA NPs had higher uptake compared with PLA and chitosan NPs after the 1st hour and increased in the next 2–4 h. However, the uptake of PLA and chitosan NPs was more evident after 4 h. The rhodamine NPs had an obvious difference in fluorescence intensity. The uptake process could be influenced by different forces, like stretching and bending force, hydrophobic forces, and electrostatic forces, or by receptor-mediated endocytoses. ζ-potential measurements were utilized to investigate the influence of carrier charge on cellular uptake. The results hypothesize that the low charge, as an absolute value, determined weaker repulsion between NPs and cell membrane (PLGA NPs −15.81) compared with the higher values (PLA NPs −30.08, chitosan NPs +34.08) [49].

### 3.2. Immortalized Cell Lines

Primary cultures of endothelial cells are used to create cell lines with extended or permanent lifespan (immortalized cell lines). They have high proliferative capacity, greater reproducibility, are cost-effective, easy to maintain in culture, and have advantages in permeability studies compared to primary cell lines. BT cell lines derived from newborn bovine nasal turbinate epithelia, do not differentiate or express tight junctions, which is why they are not useful for studying drug permeability. The most used cell lines for drug permeability studies are RMPI 2650, 16HBE14o-, Calu-3. 

#### 3.2.1. RPMI 2650 Cell Line (Human Nasal Septum Quasi Diploid Tumour Cells)

RPMI 2650 exhibited epithelial morphology from the nasal septum of a male patient with squamous cell carcinoma, a spontaneously formed tumor of the nasal septum. Its metabolic activity resembles that of normal nasal tissue. It is characterized by poor differentiation and grows to multilayers, commonly used for nasal metabolism and toxicity studies [50]. RPMI 2650 cells were initially considered unsuitable for drug transport studies, but Bai et al. found that additives and enzyme systems promoted cell differentiation and cell culture. TEER measurement was used to assess the barrier functions of epithelial cells. Each monolayer created tight junctions that largely influenced the epithelial resistance. The integrity of the nasal cell monolayer was determined by measuring the TEER values before and after drug transport [51,52]. Transwell^®^ inserts (Figure 1) are used for drug transportation studies. They are convenient, easy-to-use permeable support devices for the study of both adherence-dependent and adherence-independent cell lines. These devices are permeable and provide independent access to both sides of a monolayer. The cell line may grow under two types of culture conditions—air–liquid interface (ALI), and liquid-covered culture (LCC). Cultivation conditions—a combination of media additives plus differentiation inducers—affect the morphological and functional characteristics of the cells. In the ALI model, pre-equilibrated culture medium is added to the apical and basolateral compartments of the inserts. The upper surface of the cells is open to the air/is in contact with the air. The medium in the basal compartment is replaced regularly. Usually, a maximum TEER value is registered on the 5th day, and it can be maintained for about 10 days. The density of ciliated cells and the higher mucin secretion resemble the morphology of human nasal tissue expression [53]. In the LCC model, apical and basolateral compartments of the inserts are filled with cell culture medium. Ciliated cells are typically denuded and flattened with relatively slight mucin expression and a peak TEER value on the second day and then rapidly drop off [54]. Sibinovska et al. cultured RPMI 2650 cells under ALI and LCC conditions to investigate the permeability of 23 model drugs and several zero permeability markers. They concluded that ALI model is much more suitable than the LCC model for nasal drug permeability prediction [55,56]. 

#### 3.2.2. Calu-3 Cell Line (Human Lung Cancer Cells)

The Calu-3 cell line has characteristics equivalent to serous nasal cells, despite being derived from human lung adenocarcinoma [34]. Originally, Shen et al. showed that Calu-3 cells had a good TEER value for drug transport studies (about 100–400 Ω cm^2^). The cells form a confluent sheet and polarized monolayers with tight junctions and a uniform mucus layer [57]. Grainger et al. found that when Calu-3 cells are grown on permeable filters under ALI conditions for more than 10 days, they form differentiated layers [58], which makes them a good candidate for studying nasal drug permeation. Zhang et al. used Calu-3 cells as a nasal mucosa model to investigate the cellular-level permeability mechanism of puerarin (an isoflavone isolated from *Pueraria* root) in combination with paeoniflorin and menthol. The results confirmed that the transport of puerarin mainly occurred via passive diffusion and was increased by menthol but not by paeoniflorin [59]. Inoue et al. determined linear relationship between *in vitro* cell permeability of Calu-3 and *in vivo* bioavailability of certain drugs (antipyrine, acyclovir, caffeine, labetalol, norfloxacin, and ganciclovir) after intranasal administration in rats. This allows the quantification of drug permeation from *in vitro* drug permeability studies [60].

#### 3.2.3. Caco-2 Cell Line (Human Epithelial Colorectal Adenocarcinoma Cells)

The Caco-2 cell line was originally derived from a human colon adenocarcinoma and has been used for three decades. One of its most favorable properties is its ability to slowly differentiate into a monolayer of cells. Under normal culture conditions on semiporous filter membranes, they differentiate into enterocytes. This makes them most suitable for studying the absorption and permeability of drugs and formulations through intestinal epithelium. However, after their differentiation, they are used as a screening model to assess nasal absorption [61,62]. Among the advantages of these cells is the presence of both passive and active transport. This cell model is mainly used to assess the paracellular transport across the nasal epithelium. Unfortunately, it is unable to account for the effect of nasal mucus, mucin, clearance, and other physiological factors that impede drug permeability [62].

#### 3.2.4. Other Cell Lines

16HBE14o (16HBE) is a human bronchial epithelial cell line isolated from a 1-year-old male and immortalized with the origin-of-replication defective SV40 plasmid (pSVori-). It has been used as a model of the airway epithelium due to its morphological characteristics, barrier properties, and expression of drug transporters that are also present *in vivo* [63]. 16HBE cells can develop TEER values close to those in Calu-3 epithelial cell line [64]. 

Madin–Darby canine kidney (MDCK) cells are a model mammalian cell line used in biomedical research. They are isolated from canine distal renal tissue. When cultured on semiporous membranes, they differentiate into columnar epithelial cells and form tight junctions, such as claudin-1, claudin-4, and occludin, which lead to the formation of a restrictive paracellular barrier. They also express P-glycoprotein (P-gp) which allows them to mimic the transport across the blood–brain barrier [33,65]. The MDCK cell line can express breast cancer resistance protein (BCRP) after transfection with human ABCG2. Gonçalves et al. employed the MDCK-BCRP cell line to study bidirectional permeation of some commonly used anticonvulsants (levetiracetam, lacosamide, and zonisamide). The effect of the BCRP transporter on the rate and extent of permeation was evaluated. The obtained results confirm a successful bypass of the blood–brain barrier after intranasal administration of the used anticonvulsants [66].

There are commercially available alternative cell type models used to study drug permeation and to evaluate nasal drug delivery. The MucilAir cell line can be used to study the interaction between ATP-binding cassette (ABC) efflux transporters and intranasally administered drugs. These cells are characterized by uniform expression of cell junctions (adherents and tight junctions) joining the plasma membranes of neighboring units and forming a polarized barrier layer. Additionally, this cell line expresses various transporters like MRP1, MRP2, P-gp, and BCRP at different levels. Mercier et al. conducted bidirectional permeation studies and found that the transporters BCRP and P-gp were responsible for the efflux of the respective substrates [67]. Berger et al. cultured EpiAirway 606 in vertical diffusion chambers to study the permeation of fluticasone after intranasal administration. Various nasal spray formulations were applied to the apical surface of EpiAirway cells. The total amount of fluticasone accumulated was similar between the different formulations, but the azelastine-containing formulation of fluticasone ensured that the permeation of fluticasone occurred faster than the fluticasone-only formulation [68].

Cannabidiol (CBD)-loaded starch NPs with anti-inflammatory activity for nose-to-brain delivery were prepared by Eydelman et al. They evaluated the cellular uptake of NPs and CBD anti-inflammatory properties using the BV2 microglia cell line after 24 h treatment. The amount of cannabidiol was evaluated by HPLC on the cell surface, in the cytosol, and in the whole cell. The results showed that CBD fraction found on the cell surface was similar in all the experiments, given the concentrations of NPs were approximately the same. This confirms the fact that CBD may exhibit some of its effects through interactions with the cell membrane receptors. There were no differences between cytosol and whole-cell CBD fraction, suggesting that CBD did not cross the nuclear membrane [69].

## 4. *Ex Vivo* Methods

In science, *ex vivo* refers to experiments or measurements made in or on tissues of an organism in an external environment with minimal alteration of natural conditions. *Ex vivo* tissue-based models have been developed as an alternative to *in vivo* and cell-based models. According to biopharmaceutical drug delivery studies, the *ex vivo* model is the use of tissues that have been excised from biological individuals specifically for the purpose of experimental research [9]. Tissues can be excised from humans or animals (such as pigs, rabbits, cows, or sheep). They support the investigation of intranasal drug permeation, metabolism, efflux, and toxicity. *Ex vivo* models have limited viability which can be influenced by suboptimal storage conditions. Inter-individual differences may occur due to age, diet, and pathology and can affect tissue morphology leading to variability [33]. The time for tissue preparation in the permeability study should not exceed 4 h for maintaining maximum tissue viability. Distinct regions of the nasal mucosa can be sampled. Due to the presence of yellow pigment on the olfactory mucosa, it could be recognized to be from the pink respiratory epithelium. However, there are some limitations, including the varying thickness of the nasal epithelium among animal species and the lack of interstitial movement beneath the mucosa. On the other hand, this model is simple, time- and cost-effective, and provides reproducible results [70].

### Diffusion Chamber Devices for Ex Vivo Permeation Studies

Once the tissue is excised, it is fixed on a Franz diffusion cell (Figure 2). The apparatus consists of vertically set up donor and acceptor chambers, attached to each other. The biological membrane is set in a horizontal position in-between the two chambers with the mucosal part oriented towards the donor chamber. The diffusion medium fills the acceptor chamber which is surrounded by a water jacket that maintains body temperature within the chamber. The temperature should be around 34.4 ± 0.1 °C, the same as in the nasal cavity. “Sink” conditions should be maintained. The acceptor medium is constantly agitated by a magnetic stirring bar [71]. 

The measurement of the tissue layer thickness, usually about 200 µm, was found to be of considerable importance. In their study, Nicolazzo et al. compared the permeation of estradiol (lipophilic drug) and caffeine (hydrophilic molecule) across buccal mucosa using two types of tissues—full-thickness and epithelial tissue. Estradiol and caffeine diffusion across the epithelial tissue raised 16.7-fold and 1.8-fold, respectively, in contrast to full-thickness tissue, demonstrating the key role of the tissue thickness on the rate and extent of diffusion [72]. Also, the correct choice of acceptor medium in terms of composition has proven to be critical as it can greatly affect the results of the experiment.

At the end of the experiment the same tissue can be recovered for histological analysis. Basu and Maity evaluated the safety of their gel formulations containing Carbopol^®^ P934, sodium alginate or a combination of both for venlafaxine nasal delivery. After the fixation-staining step, the recovered tissues were compared to unexposed mucosa as a reference. The safety of the formulations was confirmed, with no visible necrosis [73]. 

Ussing vertical or horizontal chamber (Figure 2) is one of the widely used *ex vivo* models of nasal perfusion. It is simple and allows the easy maintenance of tissue viability. The permeability study can provide a quantitative assessment of passive diffusion, active transport, and efflux transport along with identification of transport routes. Different nasal mucosal efflux pumps are investigated with and without blocking agents using these models [74]. In this model, a detached epithelial membrane is placed between two chambers filled with diffusion medium and a gas mixture of 95% O_2_ and 5% CO_2_. The gas is continuously passed through the medium to preserve tissue vitality and provide appropriate hydrodynamics. Drug formulations are added to the donor chamber. Afterwards the accumulated amount of drug in the acceptor chamber is measured as a function of time. Adhesion of drug molecules to the surfaces of diffusion cells is possible [9].

Bartos et al. compared the suitability of a Franz diffusion cell model to a side-by-side/horizontal type diffusion model for different formulations (spray, gel, and powder). Although certain formulation types (spray and powder) were found to be appropriate for assessment by horizontal diffusion devices, the investigation of gel formulations appeared to be inadequate for this type of device to study diffusion processes. The Franz diffusion cell ensures an even distribution of the gel dosage form on the placed membrane. The obtained results lead the authors to conclude that vertical diffusion cells are more suitable for studying semisolid dosage forms, whereas horizontal diffusion cells are recommended for solid and liquid nasal formulations [75]. A disadvantage of vertical diffusion cells is that the donor chambers are not stirred to mimic ciliary movement of the nasal cavity to ensure even drug distribution in the donor chamber.

## 5. *In Vivo* Methods

*In vivo* models are the most accurate method to determine the pharmacokinetics of drug molecules after intranasal administration. They consider the complex interactions of various physiological factors. A thorough knowledge of the anatomy of the nasal cavity is essential before selecting an appropriate animal model. Factors such as cost, and ethics-related concerns limit animal studies. The “3Rs alternatives” (Replacement, Reduction, and Refinement) were developed to minimize the use of laboratory animals, employing precise and painless procedures [76]. The key parameters evaluated with *in vivo* models are pharmacokinetics, pharmacodynamics, and toxicological studies. A critical step in the experimental design for *in vivo* preclinical studies is the selection of appropriate animal species. It is necessary to resemble humans in terms of anatomy and physiological condition. Tentative conditions, such as sample preparation, administration, and quantification methods are of great importance for the design of a dependable *in vivo* study. There is no animal model that is morphologically and physiologically identical to human nasal characteristics. However, animal models can be very similar to humans—homogenous, resembling a human disorder—isomorphic, and can be used for the prediction of human disease and treatment [77]. Therefore, the selection of the animal model must be done cautiously. There are several physiological features of the nasal cavity and mucosa influencing drug absorption, such as surface area, ciliary movement, tight junctions, pH profile, residence time, and low enzymatic activity.

Dogs are one of the first animal models used for intranasal delivery. In early studies, dogs were used for the intranasal administration of central nervous system stimulants. Most studies reported in the literature used beagles weighing about 10 kg (1–2 years old). Beagles have large and open nostrils. They can be trained to receive medications intranasally without being anesthetized. Dogs are well-suited for clinical research; they live in environments like ours and often without a standardized diet. The administration is performed by applying 100 µL of the preparation into the nasal cavity by means of a nebulizer, or a pipette or syringe inserted about 5–10 mm into the nostril. Blood samples can be taken from the foreleg vein or hindleg vein. Despite the similarities in volume size and surface area value, the anatomy of a dog is quite different from that of a human. Nasal clearance after aerosol inhalation is completed in 2 h, indicating a clearance half-life of 30 min [10].

Guinea pigs are mostly used for immunization studies. These animals live in a standardized environment (artificial) and are on a standard diet. Dunkin disorder-isomorphic Hartley Guinea pigs weighing about 200–400 g are usually used. Their external nostrils and nasal passages are typical of mammals, but the nasopharynx is relatively short. The nasal septum contains the so-called “septal window”, so the halves cannot be treated separately. About 20–30 µL of the preparation are inserted into the nasal cavity with the help of a pipette or syringe. Blood samples can be collected from the marginal ear vein.

Hamsters are rarely used for drug delivery studies. The anatomy of the nasal cavity is similar in size and shape to that of the guinea pig [10].

Mice are easy-to-handle animals and some individuals have a short lifespan. They live in a standardized environment with a standard diet; their biochemical, physiological, and pathological criteria are known. They are usually used when they weigh about 15–20 g (4–8 weeks old) [10]. C57BL/6 mice are a commonly used inbred strain for *in vivo* studies due to their genetical identity and absence of differences that could influence experimental outcomes. They inherit all parent characteristics or phenotypes that involve appearance, behavior, or stimuli response [78].

Rabbits are preferred animals for pharmacological studies. New Zealand White and Japanese White are the most used rabbits, weighing about 2–5 kg. They are easy to handle, inexpensive and suitable for clinical studies. The anatomy of the rabbit is like that of the dog [10].

Nonhuman primate (NHP) models are primarily constructive for pathologies based on viral vector-mediated autosomal-recessive PD or neurotoxin-induced clinically familial PD. Due to their higher reproductive rate and small size, macaques and common marmosets are frequently used NHPs in the preclinical practice of PD [79,80]. Anatomically, monkeys and humans are very similar. As in humans, the olfactory region is located at the top of the nasal cavity, and it is limited to the superior nasal concha. The cilia of the olfactory mucosa are not stationary. In humans and rhesus monkeys, the olfactory region covers the same percentage of the surface area. NHPs also enable the study of neuroimaging analysis during preclinical studies of motor dysfunction that resembles the clinical PD [81]. 

### In Vivo Imaging Modalities

Imaging studies have been used to visualize drug deposition after intranasal administration in animal models. In their comprehensive review article, Veronesi et al. listed the most common *in vivo* imaging tools used in experimental testing. They emphasized the major role of these techniques in evaluating the efficacy of the therapy [82]. Optical imaging modalities such as fluorescence and bioluminescence are frequently used to provide information about the drug distribution in the body and the progression of the disease. They gauge light production for anatomical and spatial data in real-time experiments [83,84,85,86]. 

Magnetic resonance imaging (MRI) utilizes a strong magnet to produce detailed 3D images of tissues and the body. Contrast agents such as gadolinium and diethylenetriamine pentaacetic acid, are used in MRI to improve visibility of the targeted tissue. In animal models, MRI is performed prior to and after treatment to detect changes in the brain tissue as a result of the therapy. There are models for neurodegenerative disorders like PD and AD [87], neuroinflammation [88], and brain tumors [89]. Yadava et al. applied MRI imaging to demonstrate that their nanoemulsion conjugated with gadolinium was deposited in major brain regions after intranasal administration in rats. The nanoemulsion system was rich in omega-3 fatty acids from flaxseed and loaded with cyclosporine-A. The imaging results indirectly confirmed cyclosporine-A-loaded nanoemulsion reaching the same brain regions. Also, the nanoemulsion system exhibited anti-inflammatory effect in a rat model of lipopolysaccharide-induced neuroinflammation [88].

Positron emission tomography (PET) scan is an imaging technique that reveals biochemical and metabolic processes in human [90] and animal [91] tissues and organs. In PET scan, a radioactive substance called “tracer” is used to exhibit typical or atypical metabolic activity. Fluorine-18 fluorodeoxyglucose (^18^F-FDG) is one of the most widely applied tracers. By tracking the distribution of the tracers within the body, PET scans allow diagnosis and monitoring of a wide range of pathologies—dementia, AD, PD, epilepsy, cancer. Compared to MRI, PET has a lower resolution. For that reason, most PET scans are coupled with MRI systems or CT scanners. Brabazon et al. combined MRI and the ^18^F-FDG -PET system to study the effect of intranasal insulin therapy on brain glucose uptake, inflammation, learning and memory, and the volume of the lesion on adult male rats with a controlled cortical impact injury. A ^18^F-FDG uptake reduction on PET imaging and a significant decrease of lesion volume on MRI in the hippocampus were observed. Based on these data, it could be concluded that intranasally administered insulin may be used for the treatment of traumatic brain injury [92].

CT utilizes X-rays to create cross-sectional images inside selected areas of the body from different angles. It is characterized by lower tissue contrast and sensitivity of detection in comparison to PET and MRI. In nose-to-brain delivery, CT scans are mostly used to display the nasal anatomy and the dynamic airflow in animals. De Backer et al. used micro-CT image segmentation techniques to visualize the upper and lower airway morphology in Sprague–Dawley rats and to assess the deposition of the inhaled particles [93]. 

## 6. Challenges and Future Perspectives

The primary goal of research into new techniques for drug delivery to the central nervous system is to improve therapeutic outcomes. The acquisition of relevant data and translation of experimental design are considered major challenges in nose-to-brain drug delivery testing. Despite evidence for the employment of trigeminal and olfactory pathways for the direct access of bioactive molecules to the brain, some limitations may impede drug uptake by the CNS, resulting in low local drug concentrations. Computational methods, which are mainly used to investigate pharmacokinetic characteristics of drugs delivered through the nasal cavity, face a very important challenge with predicting the site and extent of drug deposition in the nasal cavity. In contrast, *in vivo* techniques allow precise delivery of the formulation to the desired area of the nasal cavity. However, interspecies anatomical and physiological peculiarities must be considered before extrapolating animal data to humans. Furthermore, nose-to-brain delivery screening tests have been shown to vary greatly between different animal species in terms of mucociliary clearance, enzymatic activity, olfactory surface area, etc. *In vitro* and *ex vivo* techniques provide certain opportunities to overcome the above-mentioned disadvantages. Cell- and tissue-based models are commonly applied in drug transport studies, but significant variations due to different expression levels of transporters or lack of a ciliated mucosal layer should not be excluded. Another significant challenge with nose-to-brain delivery is determining the drug concentration that has reached the target CNS region. Drug delivery to the brain is a function of drug properties, characteristics of the delivery system, administration technique, and experimental settings, e.g., animal species or type of disease. Scientists use different methods to quantify drug concentration in the CNS and in peripheral tissues. Drug concentration is usually determined in a tissue homogenate by UV/VIS spectrophotometric signals, fluorescence, or radioactivity. Imaging modalities can be used as another option for drug quantification, for example *in vivo* imaging of experimental animals or fluorescent microscopy of tissue slices. Experimental procedures should be described in detail and standardized reports of experimental results are needed to reliably quantify intranasal drug delivery to the brain via various possible pathways (olfactory or trigeminal nerve pathways, or absorption into the systemic circulation). It is the lack of standardization of assays and reported quantitative data from studies that confound the interpretation of experimental results. To date, no nose-to-brain research technique has proven to be exclusively applied. In order to obtain the most relevant results from preclinical studies, two or more of the described methods are recommended. Future possibilities may include 3D bio-printing, which has become one of the most advanced methods for tissue engineering. With continuous advances in the field, nasal tissues or even the entire organ designed to meet the specific geometric and functional requirements can be bio-printed, avoiding animal testing, and offering personalized therapy. Although 3D bioprinting appears to be a suitable approach for reproducing nasal cavity structures and studying nasal function, further research is needed to optimize the printing technique, bioinks, and mechanical properties to advance this promising technology. It is believed that with the efforts of researchers, a standardized 3D bio-printed model for nose-to-brain research will be produced in the future.

## 7. Conclusions

Nose-to-brain drug delivery offers many benefits over conventional routes of administration. To support computational findings with clinical value, various *in silico* and *in vitro* studies and models are progressively being used in drug discovery and development. The computational models of the nasal cavity have brought about major improvements to summarize the main characteristics of the nasal cavity. They give the opportunity to assay the factors influencing drug carrier deposition after nasal administration, such as the area of the nasal cavity, type of administration device, position of the head, angle of administration, and inspiratory flow rate, among others. Pharmaceutical and physiological factors can also affect drug permeation across nasal mucosa, and this can be assessed by *in vitro* and *ex vivo* intranasal models. They could also be used for studying the mechanism of drug absorption through the nasal epithelium. Studies with excised tissues or cultured nasal cells are useful tools to obtain information on the effect of excipients on the flux and the concentration dependence of this effect, the impact on the rate of cilia oscillation, and to find the mechanism of tissue damage by histological examination. However, by means of these models it is more important to understand the transport mechanisms across the nasal mucosa, not only if the drug is absorbed or not. Based on that, different strategies can also be developed to improve mucosal drug absorption. Even if plenty of information can be gathered with computational, *in vitro*, and *ex vivo* experiments, *in vivo* studies cannot be avoided as they are the only way to properly evaluate the behavior of a formulation after it has been administered in patients. The development and implementation of cost-effective pharmacokinetic models for intranasal drug delivery with good *in vitro*–*in vivo* correlation can accelerate the development of pharmaceutical drug products.

## Figures and Tables

**Figure 1 biomedicines-11-02198-f001:**
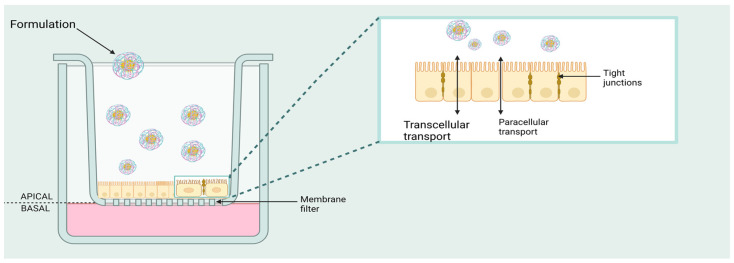
Illustration of transepithelial transport pathway of nanocarriers using Transwell^®^ inserts (ALI conditions). Transport through the cell monolayer occurs either by a paracellular or a transcellular route. Created with BioRender.com (accessed on 18 June 2023).

**Figure 2 biomedicines-11-02198-f002:**
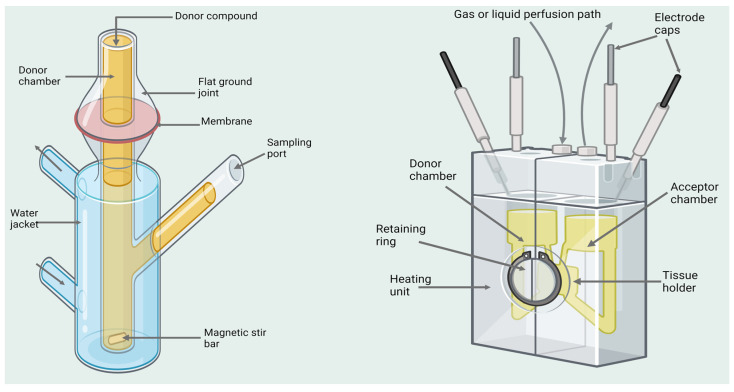
Illustration of vertical Franz diffusion cell (**left**) and Ussing vertical chamber (**right**). Created with BioRender.com (accessed on 18 June 2023).

**Table 1 biomedicines-11-02198-t001:** Advantages and disadvantages of *in silico*, *in vitro*, and *in vivo* models for evaluating nose-to-brain delivery.

Model	Advantages	Disadvantages
** *In silico* ** **(computational)**	can predict drug/carrier deposition, absorption high throughput at a reasonable cost usually are based on existing human data, so their predictions are directly applicable to humans	needs to apply several models to predict the impact of one component at a time
** *In vitro* **	no ethical considerations low-cost relative to *in vivo* control of experiment conditions feasibility of transport mechanism study	lack of actual anatomy or physiology of the nasal mucosa large intra- and interlaboratory variability lack interindividual differences
** *Ex vivo* **	maintain the integrity of the nasal mucosa availability of human nasal mucosal segments	uncontrolled experimental conditions tissue viability static system lacking blood supply
** *In vivo* **	intact physiological processes and disease featuresgold standard in preclinical phases	ethical considerations species differences between humans and experimental animals time-consuming and expensive

**Table 2 biomedicines-11-02198-t002:** Examples of *in vitro* and *in vivo* models for evaluating nose-to-brain delivery.

Type of Formulation (Drug)	Targeted Disease	*In Vitro* Model	*In Vivo* Model	Ref.
PLGA NPs (eletriptan)	Migraine	Caco-2 cell line	Wistar Albino rats	[37]
Nanosuspension (efavirenz)	Neuro-AIDS	Goat nasal mucosa	Wistar rats	[38]
PLGA NPs (levodopa)	PD	PC-12 neural-like cells	CD57/BL6 mice	[39]
Albumin NPs (R-flurbiprofen)	AD	CHO-APP695 (Chinese hamster ovary cells transfected with mouse Aβ precursor protein 695)-AD cell model	C57BL/6 mice	[40]
Solid lipid NPs (buspirone)	Anxiety disorder	Sheep nasal mucosa	Wistar Albino rats	[41]
Chitosan NPs (pramipexole)	PD	Goat nasal mucosa	Sprague–Dawley rats	[42]
Nanostructured lipid carriers, nano emulsion (escitalopram, paroxetine)	Depression	RPMI 2650	CD-1 mice	[43]
PLGA NPs (lamotrigine)	Neuropathic pain, Epilepsy	Neuro-2a cell line, RAW murine macrophage cell lines	Sprague–Dawley rats	[44]
Chitosan lipid NPs (risperidone)	Schizophrenia	Porcine nasal mucosa	Albino rat	[45]
PLGA/chitosan NPs (alpha-cyano-4-hydroxycinnamic acid, cetuximab)	Glioblastoma	Porcine nasal mucosa	Wistar rats	[46]

Abbreviations: NPs (nanoparticles), PLGA (Poly (lactic-co-glycolic acid).

## Data Availability

Not applicable.

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
