# Peer review of "Computational, In Vitro, and In Vivo Models for Nose-to-Brain Drug Delivery Studies"

_biomedicines, 2023, doi:10.3390/biomedicines11082198_

Round 1
Reviewer 1 Report
This work refers to computational, in vitro, and in vivo models for nose-to-brain 2 drug delivery studies. It is a very interesting review that covers all aspects. Some points to consider:
1. Table 1 regarding the advantages and disadvantages of in silico, in vitro, and in vivo models for evaluating nose-to-brain delivery is very informative can authors try to place it all in one page? Also, the same for table 2.
2. Please use italics thought the text when using the words: in silico, in vitro, ex vivo, and in vivo.
Author Response
We appreciate the reviewer valuable comments. The responses are provided below:
Q1: Table 1 regarding the advantages and disadvantages of in silico, in vitro, and in vivo models for evaluating nose-to-brain delivery is very informative can authors try to place it all in one page? Also, the same for table 2.
R1: Final formatting will be done after the editors accept all the corrections.
Q2: Please use italics thought the text when using the words: in silico, in vitro, ex vivo, and in vivo
R2: It is corrected.
Reviewer 2 Report
Review article is focus on the nose to brain drug delivery related model to predict drug absorption and transportation.It is good topic but need more detailing.
Accept after below correction.
1. Author very briefly highlighted different model. Write down the importance of the review in abstract.
2. Highlights the what is gap of current research ?
3. Figure 1 no meanings, in place add graphical illustration of different model.
4. Add more detail on cell line not just focus on TEER values, theirs are article where the collect drug from the cell well and measure using hplc method. Even cell imagination based article also available. Add it.
5. add more on in vivo studies based on animal imagination studies
6. Write opportunity and challenges related nose to brain model.
7. Add more recent reference, so many articles publishing related to nose to Brain model.
Author Response
We appreciate the reviewer valuable comments. The responses are provided below:
Q1: Author very briefly highlighted different model. Write down the importance of the review in abstract.
R1: Information was added in the abstract.
Q2: Highlights the what is gap of current research ?
R2: Information was added in Chapter 6 Challenges and perspectives.
Q3: Figure 1 no meanings, in place add graphical illustration of different model.
R3: We totally respect the reviewer’s opinion. With Figure 1 we aimed to illustrate the cell-based in vitro model which is commonly used to study drug transport through epithelium. We find the figure quite informative, that is why we would like to keep it in the paper.
Q4: Add more detail on cell line not just focus on TEER values, theirs are article where the collect drug from the cell well and measure using hplc method. Even cell imagination based article also available. Add it.
R4: In chapter 3.2 information regarding the most common cell lines used in transport studies is provided. The cell lines are selected based on their ability to form a confluent monolayer, which is proven by TEER measurement. That is why we have emphasized this part of the experiment.
As for the “cell imagination based…”, we are not completely sure if you understood the reviewer’s remark. We suppose that the reviewer meant imaging techniques and methods, so we have provided a separate chapter 5.1.
Q5: add more on in vivo studies based on animal imagination studies
R5: Information was added in chapter 5.1.
Q6: Write opportunity and challenges related nose to brain model.
R6: Challenges and future perspectives section was added.
Q7: Add more recent reference, so many articles publishing related to nose to Brain model.
R7: New references were added.
Reviewer 3 Report
The present review manuscript titled “Computational, in vitro, and in vivo models for nose-to-brain drug delivery studies by Boyuklieva et al. is novel and well written. In this manuscript, the authors discussed different in vitro as well as in vivo models for investigating the nose to brain drug delivery. Overall, the quality of the manuscript is good. However, the following comments need to be addressed before publication of the manuscript.
Comment 1. The role of discussed in vitro and in vivo models in the quantification of drugs delivered to the brain via the nose should be discussed in a separate section.
Comment 2. What are the challenges associated with these models? Kindly discuss.
Comment 3. Kindly discuss future perspectives for better and more accurate estimation of drugs delivered to the brain via the nose.
Author Response
We appreciate the reviewer valuable comments. The responses are provided below:
Comment 1. The role of discussed in vitro and in vivo models in the quantification of drugs delivered to the brain via the nose should be discussed in a separate section.
Response 1: Information was added in chapter 6.
Comment 2. What are the challenges associated with these models? Kindly discuss.
Response 2: A new chapter was added.
Comment 3. Kindly discuss future perspectives for better and more accurate estimation of drugs delivered to the brain via the nose.
Response 3: Information is provided in chapter 6.
Round 2
Reviewer 3 Report
The authors revised the manuscript very critically. I don't have further comments.